# Intercomparison of commercial analyzers for atmospheric ethane and methane observations

Róisín Commane[1,2], Andrew Hallward-Driemeier[2], Lee T. Murray[3,4]

[1]Department of Earth and Environmental Sciences, Columbia University, New York, NY 10027, USA
[2]Lamont-Doherty Earth Observatory, Columbia University, Palisades, NY 10964
[3] Department of Earth and Environmental Sciences, University of Rochester, Rochester, NY 14627, USA
[4] Department of Physics and Astronomy, University of Rochester, Rochester, NY 14627, USA

*Correspondence to*: Roisin Commane (r.commane@columbia.edu)

**Abstract**

Methane ($CH_4$) is a strong greenhouse gas that has become the focus of climate mitigation policies in recent years. Ethane/methane ratios can be used to identify and partition the different sources of methane, especially in areas with natural gas mixed with biogenic methane emissions, such as cities. We assessed the precision, accuracy and selectivity of three commercially available laser-based analyzers that have been marketed as measuring instantaneous dry mole fractions of methane and ethane in ambient air. The Aerodyne SuperDUAL instrument performed best of the three instruments but it is large and requires expertise to operate. The Aeris Mira Ultra LDS analyzer also performed well for the price point and small size but required characterization of the water vapor dependence of reported concentrations and careful setup for use. The Picarro G2210-i precisely measured methane but it did not detect the 10 ppbv (part-per-billion by volume) increases in ambient ethane detected by the other two instruments when sampling a plume of incompletely combusted natural gas. For long-term tower deployments or those with large mobile laboratories, the Aerodyne SuperDUAL provides the best precision for methane and ethane. The more compact Aeris MIRA can, with careful use, quantify thermogenic methane sources to sufficient precision for mobile and short-term deployments in urban or oil and gas areas. We weighed the advantages of each instrument, including size, power requirement, ease of use on mobile platforms, and expertise needed to operate the instrument. We recommend the Aerodyne SuperDUAL or the Aeris MIRA Ultra LDS depending on the situation.

## 1 Introduction

The atmospheric concentrations of methane ($CH_4$), a strong greenhouse gas, have been rising at an unprecedented rate in recent years, with record breaking growth rates since 2020 (https://gml.noaa.gov/ccgg/trends_ch4/). Methane has an atmospheric lifetime of ~10 years compared to ~100 years for carbon dioxide ($CO_2$) and absorbs over 80 times more heat than $CO_2$ over 20 years (Szopa et al., 2021). Both of these characteristics make the reduction of methane emissions a priority target for short-term reductions in anthropogenic global warming. In recent years, methane has become the target of climate mitigation policies at many levels of government, including international (e.g. founding of the United Nations Environment Programme funded International Methane Emissions Observatory (IMEO) in 2022), national (e.g. Inflation Reduction Act, 2022, USA) and local (e.g. New York's Climate Leadership and Community Protection Act, (CLCPA), over 50 cities in the US banning natural gas new construction).

Methane sources are categorized as thermogenic (e.g. oil, natural gas, coal mining) or biogenic; which can be both natural (e.g. wetlands) or anthropogenic (e.g. agriculture, landfills, sewage) in origin (Saunois et al., 2020). Each of these methane sources co-emits different trace gas species, which we can use to identify the source of methane. Thermogenic sources of methane, such as natural gas, also contain ethane ($C_2H_6$) and other hydrocarbons. The incomplete combustion of liquid (e.g. natural gas) or solid (e.g. coal, wood) fuels can co-emit high concentrations of carbon monoxide (CO) and other Volatile Organic Compounds VOCs. Biogenic sources of methane do not co-emit ethane, but can emit carbon dioxide ($CO_2$) and more odorous trace gases such as hydrogen sulfide ($H_2S$). Therefore, ethane can be used to distinguish between thermogenic (methane/ethane co-emitted) and biogenic (no ethane emitted) sources of methane. Many studies have used methane/ethane ratios to identify natural gas leaks in the natural gas production and distribution networks (Smith et al., 2015; Wunch et al., 2016; Gvakharia et al., 2017; Floerchinger et al., 2019). Methane/ethane observations have also been used for mobile and stationary sampling in urban areas across many countries to identify natural gas leaks separately from biogenically produced methane (McKain et al., 2015; Lamb et al., 2016; Maazallahi et al., 2020; Defratyka et al., 2021).

Methane monitoring networks are being developed for city, state and national scales with the goal of evaluating the efficacy of methane reduction policies (Karion et al., 2020; Sargent et al., 2021; He et al., 2019; Mueller et al., 2021). Many of these networks will need to partition the contribution of methane between thermogenic and biogenic sources. In recent years, commercial analyzers have been developed to measure methane and ethane at ambient concentrations and many of these analyzers are marketed as allowing users to attribute the sources of methane. As far as we can tell, there has not yet been a systematic assessment and characterization of these newly available laser-based ethane spectrometers. There is also little guidance available to those now charged with instrumenting networks and mobile platforms for methane source apportionment.

Here, we evaluated three laser-based spectrometers that are marketed to measure ambient dry mole fractions of ethane and methane; (i) a cavity enhanced infra-red (IR) absorption spectrometer (Aerodyne Research Inc SuperDual QCl/ICL), (ii) a mid-IR absorption spectrometer (Aeris Technologies Mira Ultra LDS) and (iii) a cavity ring down spectrometer (Picarro G2210-i CRDS). The precision and accuracy of each instrument was evaluated and compared to the advertised performance. We tested the water vapor response and assessed the long-term operation needs of each instrument. Finally, we evaluated the performance of each instrument while sampling urban air at a rooftop site with large natural gas and biogenic emissions in the urban core of New York City in February 2022. We examine the requirements for long-term operation of each analyzer and make recommendations for operation.

## 2 Methods

2.1 Description of Analyzers

Each of the analyzers described below reports the dry mole fraction of methane and ethane in air using units of ppbv, parts-per-billion by volume, which is the equivalent of nmol mol$^{-1}$ for an ideal gas.

2.1.1 Aerodyne Research Inc SuperDual

Various configurations of Aerodyne laser spectrometers have been used to measure methane and ethane in stationary (McKain et al., 2015), ground-based mobile (Yacovitch et al., 2014), and airborne (Kostinek et al., 2019; Plant et al., 2019) platforms. These spectrometers use a continuous wave interband cascade laser (ICL) based spectrometer to measure methane, ethane and water vapor. ICLs are often used in a two laser system alongside a continuous wave quantum cascade laser (QCL) to measure dry mole fractions of carbon dioxide ($CO_2$), carbon monoxide (CO), and nitrous oxide ($N_2O$). Here, we use a SuperDUAL configuration of a two-laser system with a 2L astigmatic Herriott cell (path length 210m) at 50 Torr pressure. The instrument was manufactured in 2015 and refurbished with new lasers in 2020. We use the provided TDLWintel software to fit the absorption spectra and quantify five target gasses and water vapor. The ICL (Laser 1) sweeps from 2988.520 to 2990.625 cm$^{-1}$ to detect $CH_4$, $C_2H_6$ and $H_2O$. The edge of the ethane absorption feature (2990.081 cm$^{-1}$) includes a small methane peak (2989.98 cm$^{-1}$) that is fixed to the value determined from the main fit at 2989.003 cm$^{-1}$. The QCL (Laser 2) sweeps from 2227.550 to 2228.000 cm$^{-1}$ and includes absorption features for $^{13}CO_2$ (2227.605 cm$^{-1}$), CO (2227.639 cm$^{-1}$), $N_2O$ (2227.843 cm$^{-1}$) and $H_2O$. We use the default water broadening coefficient (WBC) for all species (WBC = 2) except CO (WBC = 1.45). The analyzer is large and heavy (56 cm x 77 cm x 64 cm; 75kg) and requires an external pump and chiller (to maintain laser temperature stability) that require a stable power source. The instrument has been used extensively and successfully for long-term ground site observations and mobile lab deployments but it is not suitable for smaller/car based mobile sampling. As part of our regular ambient sampling, the Aerodyne SuperDUAL samples nitrogen gas each hour to account for instrument drift, which is especially evident in lower concentration species such as ethane. A smooth spline is fitted to the reported zero for each gas species and subtracted from the 1Hz data.

2.1.2 Aeris Technologies MIRA Ultra LDS

The Aeris Technologies MIRA Ultra LDS (#100209; manufactured July 2021) uses a mid-IR ICL (~3000 cm$^{-1}$ range) with a multi-pass cell. There are few descriptions of the Aeris MIRA but (Travis et al., 2020) described a similar, portable version of the instrument with an onboard battery (MIRA Pico, not evaluated here). The multi-pass cell (60 cm$^3$) has a path length of 13 m and an internal pump maintains the cell pressure at 180 Torr with a ~380 sccm flow rate. The small footprint of the rackmount configured analyzer (43 cm x 28 cm x 13 cm; 5 kg) makes it ideal for car-based mobile sampling. The current configuration using a small internal pump is not suitable for sampling below ambient pressure and care should be taken when configuring the system when sampling through long lines on towers.

2.1.3 Picarro G2210-i

The Picarro G2210-i (#3441-RFIDS2010, manufactured Aug 2019) is a Cavity Ring Down Spectrometer that measures $CH_4$, $CO_2$, $C_2H_6$, and $\delta^{13}C\text{-}CH_4$. The instrument uses an external pump to reach a cell pressure of 148 Torr and flow rate of 24 sccm through a cavity of 35 cm$^3$ with a path length of up to 30 km

(https://www.picarro.com/support/library/documents/g2210i_analyzer_datasheet). The measurement and reporting
cycle of the Picarro G2210-i are 1Hz. But the low flow rate reduces the instrument response time considerably. We
have corrected for the delay and report methane at 1Hz and we have averaged the ethane to 10s and 5 minutes. Methane
data from the instrument has been used on mobile (O'Connell et al., 2019) and stationary (Lebel et al., 2020) platforms
and is also mentioned in (Defratyka et al., 2021) but none of these studies have discussed or shown the observed
ethane concentrations. The datasheet indicates that the instrument is designed to sample ambient air but may have
interferences from elevated concentrations of gas species such as hydrogen sulfide ($H_2S$) or volatile organic
compounds (VOCs).

### 2.2 Instrument Evaluation Set-up

#### 2.2.1 Humidity

The humidity of the sample line for the instruments was varied using a Perma Pure Nafion (™) dryer. Nafion dryers
have a semi-permeable membrane separating an internal sample gas stream from a counterflow purge gas stream
contained within a stainless-steel outer shell. If the partial pressure of water vapor is higher in the purge gas stream,
then water is added to the sample gas stream. A counter flow of air was drawn through the Nafion at ~2000 sccm
using a vacuum pump. The inlet to the counter flow was alternatively sampling the top of a container of water or dry
air-conditioned air in the observatory. To achieve the lowest humidity, dry nitrogen was pushed through the Nafion.
The flow rate through the Nafion was controlled using a ball valve and allowed for different rates of changes in the
humidity. No liquid water was introduced to the sample lines for the instruments. A range of water vapor from 3% to
0.05% was used for all instruments except for the Aeris Mira Ultra LDS ethane data, which was cut off at 1.05% water
vapor (for reasons discussed below).

#### 2.2.2 Calibrations Against NOAA Standards

Each of the instruments sampled two ambient range cylinders calibrated by the Central Calibration Laboratory (CCL)
at the National Oceanographic and Atmospheric Administration (NOAA) Global Monitoring Laboratory (GML) in
Boulder, CO. CCL maintains the World Meteorological Organization (WMO) methane scale (WMO X2004A) and an
internal CCL standard for ethane (C2H6-2012). A dry, compressed air cylinder was used to test multi-hour instrument
stability.

#### 2.2.3 Instrument Precision

We evaluated the instrument precision by running a calibrated compressed air cylinder for a 4 hour period and
calculating Allan-Werle variance and precision (also called continuous measurement repeatability (CMR) (Defratyka
et al., 2021; Yver Kwok et al., 2015). During this time the regular zero for the Aerodyne SuperDUAL was not
performed to allow for direct comparison of all instruments. The Aeris MIRA and Picarro G2210-i were humidified
(1.7 - 1.9 % $H_2O$) to allow the Aeris MIRA to report ethane (see Section 3.1). The Aerodyne SuperDUAL was not
humidified and reported less than 0.054 % $H_2O$ for the same tests.

 2.2.4 Nitrogen Tests

During regular ambient operation, the Aerodyne SuperDUAL samples nitrogen gas each hour to account for
instrument drift. We use the boil off from a large liquid nitrogen dewar, which can be refilled on site, and which
contains a variable mole fraction of carbon monoxide (~250 ppbv), and may contain trace levels of oxygen and argon.
Regular nitrogen sampling is not required for the long-term operation of either the Picarro G2210-i or the Aeris MIRA.
We evaluated the short-term repeatability of the Aeris MIRA and Picarro G2210-i when sampling dry and humidified
nitrogen.
2.3 Site Description and Sampling of Ambient Urban Air
The City University of New York (CUNY) Next Generation Environmental Sensor (NGENS) Observatory is on the
rooftop of the 56m building in Hamilton Heights in Harlem. The sampling point is ~93m above sea level on a tower
at the south end of the building. The Aerodyne SuperDUAL has been operated at the site over a number of years and
was running from early January - June 2022. The site samples urban air that has been influenced by natural gas
emissions (both pre and post combustion), wastewater treatment plants (North River to the north-west, Ward Island
to the east) and sewer street emissions. During the long-term operation of the Aerodyne SuperDual, nitrogen (liquid
nitrogen boil off, $N_2$) is added as a test of the zero drift in the instrument. For the experiments described here, $N_2$ was
used hourly during ambient sampling and prior to and after the compressed air tank test runs. When the Aerodyne
SuperDUAL is operated independently, air is drawn through ~10 m of ½" Synflex tubing at 20 L min$^{-1}$ using a
diaphragm pump before being sub-sampled by the Aerodyne SuperDUAL (flow rate 1.7 L min$^{-1}$). The use of a separate
pump to increase the total flow rate and reduce instrument response times is commonly used for ground operation
with longer tubing (e.g. towers). However, the pump also reduces the pressure within the tubing to below ambient
pressure, which was a problem when sampling with the smaller pump capacity of the Aeris MIRA. For the work
described here, the external pump was removed and the response time through the tubing was reduced to 30s. Each
instrument sampled from a Swagelok cross fitting using a ~1m ¼" Synflex tubing.
We sampled air from the roof in February, 2022 when ambient air temperatures ranged from below freezing
(-9.3°C) to a warm spring day (19°C). The lowest temperatures were also associated with low humidity, which caused
problems that were also detected during the humidity testing, so the sample line of the Picarro G2210-i and Aeris
MIRA were humidified to >1% water vapor as a work around for these problems.
**3 Results and Discussion**
We characterized the laboratory performance of each analyzer with respect to humidity corrections, precision
assessment, calibration to NOAA standards and long-term stability, before sampling ambient air in New York City.
We used these tests to recommend the best instrument for use in different circumstances.
3.1 Characterization of Water Sensitivity
All three instruments showed a dependency on water vapor for methane that was statistically significant. Figure 1
shows the dependence of the retrieved methane and ethane with the water vapor reported by each instrument for a
compressed air cylinder with variable humidity. A linear correction was calculated for methane and ethane for both
the Aerodyne SuperDUAL and the Picarro G2210-i but a quadratic dependence was observed for the Aeris MIRA
methane (Fig S1). The values of each water vapor correction are shown in Table 1. The Picarro G2210-i needed the
smallest absolute correction for methane, and the Aerodyne SuperDUAL reported the smallest correction for ethane.
The SuperDUAL was operated with the default water vapor broadening coefficient for methane and ethane of 2.0.
This correction is likely too large for methane and moving closer to the value of 1.05 recommended by Kostinek et
al., 2019 would reduce the water vapor correction. Here we have applied a linear correction with water vapor to the
observed data that results in a 10 ppbv change in methane but a ~0.08 ppbv change in ethane for 0-2% water vapor.

**Table 1: Summary of water vapor corrections derived for each instrument.**

| Instrument | $CH_4$ Correction ppbv/% $H_2O$ <br> y = m * [$H_2O$] | $C_2H_6$ ppbv/% $H_2O$ <br> y = m * [$H_2O$] | Notes: Using default water broadening coefficients for all instruments before calibration |
|---|---|---|---|
| Aerodyne SuperDUAL | -5.335 ppbv /% $H_2O$ | 0.042 ppbv /% $H_2O$ | |
| Aeris MIRA Ultra LDS | -25.53 (ppbv /% $H_2O$)$^2$ - 59.22 ppbv /% $H_2O$ | 0.23 ppbv / % $H_2O$ | $C_2H_6$ only calculated for $H_2O$ > 1.05 % |
| Picarro G2210-i | -1.15 ppbv /% $H_2O$ | -0.82 ppbv /% $H_2O$ | |

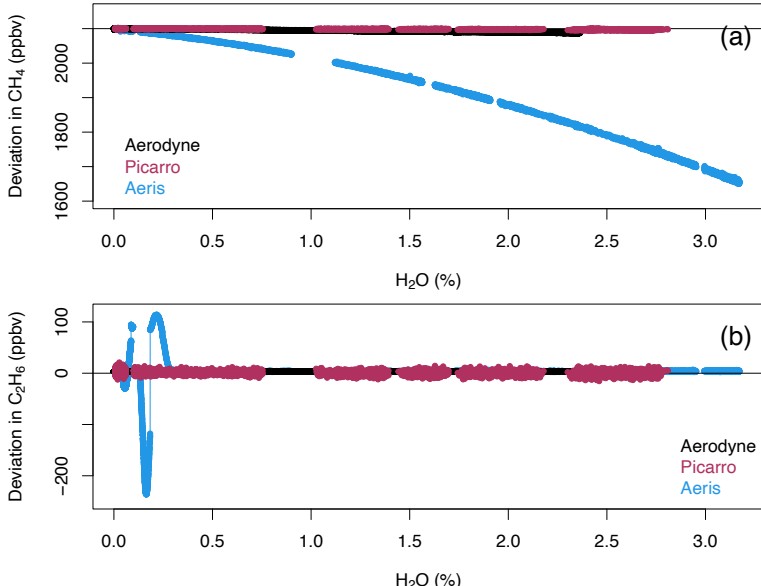


**Figure 1. Uncorrected (a) methane (ppbv) and (b) ethane (ppbv) vs water vapor (%) for the Aerodyne SuperDUAL**
**(black), Picarro G2210-i (red) and Aeris MIRA Ultra LDS (blue).**

We identified two separate, but related, situations with the Aeris MIRA that could prove to be a problem if not
accounted for in operation in certain environments and configurations:
(i)      The wavelength of the laser is tied to the water vapor absorption peak. When running a dry calibration tank,

the instrument loses frequency lock and the laser wavelength can drift to the point that the ethane peak can

no longer be resolved. The reported ethane concentrations vary between 200 ppbv and -100 ppbv during this

dry air sampling, possibly driven by laser wavelength drift. When the water vapor increases again after a

calibration, the ethane fit is not immediately recaptured. Noise in the reported ethane and methane

concentrations increases significantly below 1.05% water vapor and below 0.5% the ethane fit is completely

lost. After discussion with engineers at Aeris Technologies, we learned that there are two water vapor peaks

| 198 | in the spectral window. This problem could be mitigated when sampling dry cylinders by locking to the |
| 199 | stronger water vapor absorption peak, which is often saturated during normal operation, or to the methane |
| 200 | line directly. Note that locking to the methane line would prevent running zero methane or nitrogen samples |
| 201 | as discussed in Section 3.4 below. Either change can be implemented upon request when ordering new |
| 202 | analyzers. |
| 203 | (ii) | For most environments, water vapor in the atmosphere absorbs some of the mid-IR laser power and the laser |
| 204 | power of the Aeris MIRA is optimized to achieve maximum sensitivity. However, New York City in |
| 205 | February is cold and dry, with very low concentrations of ambient water vapor. Without enough water vapor |
| 206 | to attenuate the laser power, the detector can be saturated, leading to no ethane detected and a noisy methane |
| 207 | retrieval. This problem can be fixed by reducing the laser power slightly (using the procedure recommended |
| 208 | by Aeris Technologies, personal communication) or by humidifying the sample line slightly. We opted for |
| 209 | the latter fix for this study. At the other extreme, water vapor closer to 3% can also lead to increased noise in |
| 210 | the fitted methane and ethane. |
| 211 | After losing the ethane peak during either of these circumstances, the Aeris MIRA analyzer will often fail to find the |
| 212 | peak again until manually re-connected to the internet. We have not identified a cause for this behavior but it was |
| 213 | more likely during (ii) and was not a problem after we humidified the sample flow. Using the GPS receiver provided |
| 214 | by Aeris also seemed to mitigate the problem. |

3.2 Instrument Calibration
Each instrument was calibrated against two NOAA calibration standards after accounting for the water vapor
correction described in Section 3.1. A linear fit (OLS, ordinary least squares) was calculated for each species and the
span (slope) and zero correction (intercept) and 95% confidence intervals were calculated (Table 2). The span and
offset were then applied to each species. As described above, the Aeris MIRA could not report ethane concentrations
when sampling a dry tank so the sample line of both the Aeris MIRA and Picarro G2210-i were humidified to water
vapor mole fractions between 1.7-1.9 % $H_2O$. For methane, all three instruments reported a span correction less than
3%, and zero corrections between 3 and 14 ppbv. All three instruments report very similar methane mole fractions for
a compressed air tank after all calibration steps were applied. For ethane, the Aeris MIRA and Aerodyne SuperDUAL
reported a span less than 7% and offset of less than 2 ppbv. However, the slope and intercept for the Picarro G2210-i
were not successfully resolved for the reported 1 Hz data and a two-point linear fit was calculated for the average
values reported over the sampling period (Slope 0.427; intercept 4.275). The resulting correction successfully resolved
the target gas mole fractions but with a large standard deviation in the 1 Hz data (Fig S2).
**Table 2. Calibration span (slope) and zero (intercept) calculated for each instrument reporting at 1 Hz when sampling the**
**NOAA calibration standards. The 95% confidence intervals (CI) for the slope and intercept of an Ordinary Least Squares**
**(OLS) fit are also shown. **The ethane Picarro G2210-i calibration was calculated from the mean of each cylinder**
**measurement (two-point calibration).**

| Species | Slope | Intercept | 95% CI Slope +/- | 95% CI Intercept +/- | $r^2$ |
|---|---|---|---|---|---|
| Aeris; $CH_4$ (ppbv) | 0.977 | -4.2 | 0 | 0.4 | 1 |
| Aeris; $C_2H_6$ (ppbv) | 0.992 | -2.42 | 0.01 | 0.07 | 0.9806 |

| | | | | | |
|---|---|---|---|---|---|
| Picarro; $CH_4$ (ppbv) | 1.002 | 1.4 | 0 | 0.5 | 1 |
| Picarro; $C_2H_6$ (ppbv)** | 0.42 | 4.28 | | | |
| Aerodyne; $CH_4$ (ppbv) | 0.969 | -13.9 | 0.001 | 0.2 | 1 |
| Aerodyne; $C_2H_6$ (ppbv) | 1.069 | 0.064 | 0.001 | 0.004 | 0.9996 |


We evaluated the linearity of the instruments outside our range of calibration standards by comparing the instrument
response for the Aerodyne SuperDUAL and Aeris MIRA during the high plumes (as discussed in Section 3.5 below).
Fig S3 shows the linearity of 1s methane and 10s ethane for Feb 20-21, 2022 with the Aeris MIRA and Aerodyne
SuperDUAL. The methane fit (Slope 1.002) is slightly closer to 1 than the ethane fit (Slope: 1.048 +/- 0.002). The
slow response of the Picarro G2210-i meant that it could not represent plumes of ethane at sufficient resolution to
allow for valid comparison. While this does not directly test linearity, the strong correlations between reported
concentrations from the instruments likely indicates that they retain linear behavior well beyond the range of our
calibration standards.
3.3 Instrument Precision
The precision of each analyzer was evaluated by sampling a calibrated compressed air cylinder for four hours. We
calculated an Allan-Werle variance (Fig 2) and the observed precision for methane and ethane for each instrument
(Table 3; Fig S4-S7).

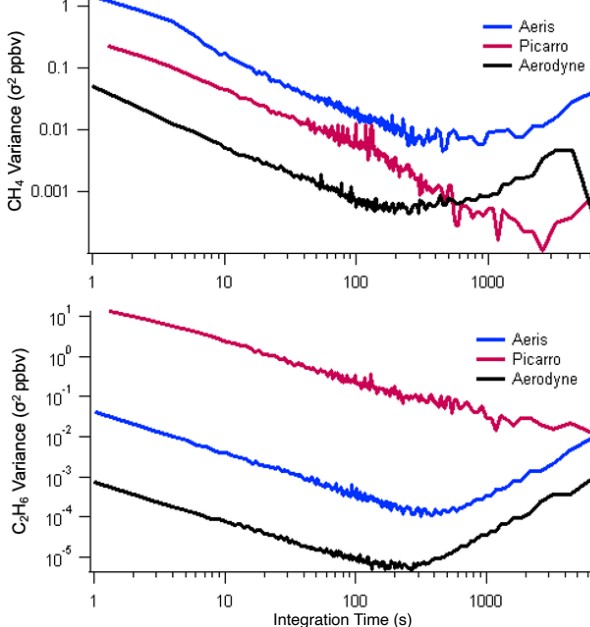

**Figure 2. Allan-Werle Variance for (a) methane and (b) ethane for all three instruments when sampling a compressed air**
**cylinder on Feb 17th, 2022 11 am - 3 pm EDT. Each of the tanks was calibrated to NOAA cylinders after water vapor**
**correction. The reported water vapor for the Aerodyne SuperDUAL (black) was below 0.054 %. The Aeris MIRA (blue)**
**and Picarro G2210-i (red) were humidified to water vapor 1.7 – 1.9 %.**

**Table 3: Summary of various instrument performance metrics. The quoted precisions are from the Product Datasheet for**
**each analyzer except *Aerodyne SuperDUAL quoted precision from Kostinek et al., 2019**

| Instrument Manufacturer | Flow Rate | $CH_4$ Quoted Precision | $CH_4$ Observed Precision (100 s) | $C_2H_6$ Quoted Precision | $C_2H_6$ Observed Precision (100 s) |
|---|---|---|---|---|---|
| Aerodyne SuperDUAL | 1500 sccm | 0.025 ppbv* (100 s) | 0.024 ppbv | 0.003 ppbv* (100s) | 0.003 ppbv |
| Aeris MIRA Ultra LDS | 380 sccm | 0.5 ppbv (1 sec) | 0.14 ppbv | 1 ppbv (1 sec) | 0.02 ppbv |
| Picarro G2210-i | 24 sccm | <0.1 ppbv (5 min) | 0.08 ppbv | <1 ppbv (5 min) | 0.48 ppbv |


For methane, the Aerodyne SuperDUAL had the best 1 Hz (0.227 ppbv) and 10s (0.072 ppbv) precision with
a minimum of 0.021 ppbv at 3.2 mins but the variance increased slightly again (but still below 1 ppbv) after about 15
mins. There were no zeros performed for the SuperDUAL during the precision experiment so this increase in variance
was not unexpected. The Aerodyne SuperDUAL matched the 100 s precision of (Kostinek et al., 2019) at 0.024 ppbv.
At 100s, the Aeris LDS precision was 0.14 ppbv and the Picarro G2210-i precision was 0.08 ppbv, both of which
exceeded their quoted precision of 0.5 ppbv (at 1 s) and 0.1 ppbv (at 5 min).
For ethane, the Aerodyne SuperDUAL had the best 1Hz (0.027 ppbv) and 10s (0.008 ppbv) precision with a
minimum of 0.002 ppbv at 2.2 mins but the variance increased slightly again (but still below 0.03 ppbv) after about
15 mins. The Aerodyne SuperDUAL matched the 100s precision of (Kostinek et al., 2019) of 0.003 ppbv. At 100s,
the Aeris MIRA precision was 0.02 ppbv and the Picarro G2210-i precision was 0.48 ppbv, both of which exceeded
their quoted precision of 1 ppbv.
3.4 Long-term Instrument Stability
We evaluated the stability of frequent additions of nitrogen (liquid nitrogen boil-off free of methane, ethane, $CO_2$,
etc.) for all three analyzers. Fig 3 shows the instrument response when sampling dry and humidified nitrogen (methane
and ethane free). The Aerodyne SuperDUAL was not humidified for the second period (Fig 3c-d) and the noise was
not significantly different for the two periods ($C_2H_6$ < 0.01 ppbv; $CH_4$ < 0.95 ppbv; 1σ s.d.).
The Aeris MIRA instrument response is statistically different when sampling dry or humidified nitrogen (Fig
3): The reported ethane goes from varying between -100 and 100 ppbv (with a mean of -3.92 ± 43.8 ppbv; 1σ s.d.)
when sampling dry nitrogen to -0.05 ± 0.22 ppbv (1σ s.d.) when the nitrogen is humidified to ~1%. This result is
consistent with the humidity test with compressed air in Figure 1. However, humidifying the nitrogen also affects the
reported methane, which goes from 0.02 ± 0.5 ppbv (1σ s.d.) when dry to 2.5 ± 17.5 ppbv (1σ s.d.) when humidified.

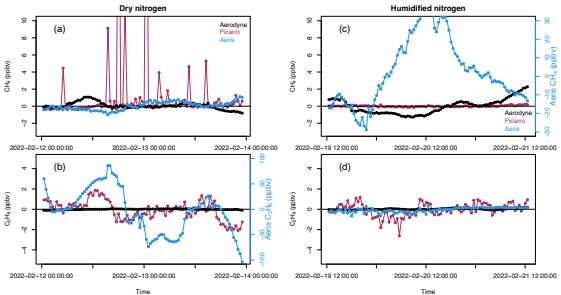

**Figure 3: Instrument response when sampling (a-b) dry and (c-d) humidified methane (a, c) and ethane (b, d) in nitrogen. Picarro G2210-i (red) and Aeris MIRA (blue). Note the separate right y-axis for the Aeris (b) ethane and (c) methane. Also note that the Aerodyne SuperDUAL (black) did not sample humidified nitrogen in c-d.**

The Picarro G2210-i instrument noise is reduced when sampling humidified nitrogen over dry nitrogen (Fig 3), especially for outliers in the reported methane (Fig 3a). The reported ethane goes from -0.082 +/-0.95 ppbv (1σ s.d.) when sampling dry nitrogen to -0.03 ± 1.73 ppbv (1σ s.d.) when the nitrogen is humidified to ~1%. The reported methane goes from 1.35 ± 6 ppbv (1σ s.d.) when dry to 0.007 ± 0.08 ppbv (1σ s.d.) when humidified.

Using a Picarro G1301, (Nara et al., 2012) observed a pressure broadening effect when sampling gas with a range of oxygen and argon that resulted in a ~2 ppb bias in methane. We would expect to see a larger pressure broadening effect when sampling dry nitrogen free of oxygen and argon, which may explain some of the variability in Fig 3a. Indeed, there is no increased variability in methane observed by the Picarro G2210-i when sampling from a compressed air cylinder at low humidity (Fig 1(a)). For the Aeris MIRA we see different behavior for the methane and ethane. The ethane results are consistent for both compressed air and nitrogen with more ethane variability at low humidity. The methane variability is much larger when sampling humidified nitrogen and dry compressed air than seen when sampling dry nitrogen and humidified compressed air (see Fig 1 and S1). In our tests here, the G2210-i stability for methane is the best of the three analyzers when sampling humidified nitrogen boil off, which indicates that the addition of nitrogen from a dewar is possible as a long-term zero only if the flow is humidified. However, for the Aeris MIRA, we observe much more methane variability in humidified nitrogen and lots of ethane variability in dry nitrogen so we do not recommend using nitrogen as a long-term zero.

3.5 Ambient Sampling

In order to test the suitability of each analyzer to report accurate methane and ethane mole fractions in ambient air, we ran all instruments sampling ambient air from the CUNY Observatory in Harlem, NY, for 3-4 weeks in February 2022. In general, air is cold and very dry in New York City in winter and and it took some time to learn that we had to humidify the Aeris MIRA and Picarro G2210-i sample flows in order to record valid data (see instrument characterization experiments described above). The Picarro G2210-i was often reporting negative ethane and negative correlations of ethane with methane for the first two weeks of observations. We then requested that Picarro engineers check the instrument and they assured us it was performing as expected. So we have focused on Feb 17-22, 2022 (see Fig S8), when the G2210-i was confirmed to be performing to specification. Figure 4 shows typical examples of the ambient methane and ethane mole fractions observed by all the analyzers when sampling ambient air in February 2022.

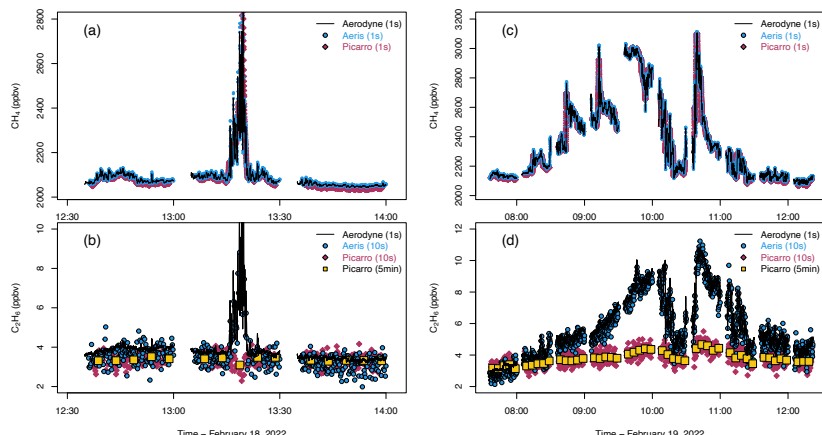

**Figure 4: Ambient sampling for methane (ppbv, top row) and ethane (ppbv, bottom row) for (a-b) a short natural gas plume on Feb 18th, 2022 and (c-d) a large scale change in methane and ethane overnight and into the early morning of February 19th, 2022. Times in UTC. Aerodyne SuperDUAL (black line), Aeris MIRA (blue circle), Picarro G2210-i (red diamond) and Picarro G2210-i averaged to 5 minutes in yellow square. All instruments were corrected for humidity and calibrated to NOAA calibration scales.**

On February 18th a large-scale change in airmass resulted in a drop in ambient air temperature from 15°C to 7°C (Fig 4a and b), residential heating increased and a plume of high methane and ethane was intercepted at the observatory for about 10 minutes. The Aerodyne SuperDUAL and Aeris MIRA both responded very similarly; reporting large coincident increases in methane (up to ~2800 ppbv) and ethane (~10 ppbv). The Aerodyne SuperDUAL also reported a large increase in carbon monoxide (CO) up to ~1500 ppbv for the same plume, possibly indicating an incomplete combustion source. The methane reported by the Picarro G2210-i also increased, but with a longer peak duration due to the much slower sampling flow rate (sampling time lags were corrected for previously). However, the ethane surprisingly decreased while sampling the plume.

On February 19th ambient air temperatures ranged from -3.7°C at night to -1.2°C in the early morning and wind speeds were low (2-4 m s$^{-1}$) leading to a build-up of methane and ethane in the atmosphere overnight (Fig 4 c and d). The prolonged elevated methane (to ~3000 ppbv) and ethane (to ~11 ppbv) was easily observed by the Aerodyne SuperDUAL and the Aeris MIRA. The CO also increased (~700 ppbv) to about half of that seen on February 18th. The methane reported by the Picarro G2210-i also increased in line with the other reported methane but, again, the Picarro G2210-i was not able to resolve the large increase in ethane, this time indicating an increase in ethane of 1-2 ppbv instead of the 7-8 ppbv seen by the other instruments.

The trace gases measured by the Aerodyne SuperDUAL indicate that Fig 4 (a and b) shows a post-meter plume of incompletely combusted natural gas, likely emitted close to the observatory. The overnight boundary build-up observed in Fig 4 c and d was coincident with a large increase in other combustion pollutants such as CO. As mentioned in the data sheet for this instrument, it is possible that the co-emitted species of natural gas combustion (such as CO or other volatile organic compounds, VOCs) are acting as an interferent for the Picarro G2210-i ethane retrieval. Our results indicate that the Picarro G2210-i should not be used to selectively measure ethane near combustion sources such as flares, or natural gas power plants or in urban areas that combust natural gas on a large scale. Indeed, care should be taken to ensure that thermogenic sources are not erroneously attributed to biogenic sources with the Picarro G2210-i in urban areas.

**4 Conclusions and Recommendations**

We evaluated the performance of three commercially available laser-based ethane analyzers: Aerodyne Inc.
SuperDUAL, Aeris Technologies MIRA LDS, Picarro Inc. G2210-i. We assessed the precision, accuracy and
interferences of each analyzer. We measured ambient air in a cold urban environment with each analyzer and have
made recommendations of analyzers based on performance, ease of use and reliability.
Across the month, the Aerodyne SuperDUAL reported with the highest precision of all three instruments but
requires regular zero air/nitrogen to maintain accuracy. The large size of the instrument and external chiller and large
pump mean that it is more suitable for tower/ground-based or large mobile laboratory operation and is not suitable for
car-based sampling. There is a smaller size instrument from Aerodyne – the Aerodyne "mini" – which has a
methane/ethane precision between that of the SuperDUAL and the Aeris MIRA but this also requires an external
chiller and large pump (see https://www.aerodyne.com/wp-content/uploads/2021/11/Ethane.pdf; 60s precision of 0.05
ppbv $CH_4$ and 0.015 ppbv $C_2H_6$). The Aerodyne SuperDUAL also requires expertise to operate and maintain but is
the best performing analyzer if the space and expertise are available.
The Aeris MIRA was close to the Aerodyne SuperDUAL for precision for methane but was less precise for
ethane. The Aeris MIRA pump is small so the analyzer cannot draw against pressures much below ambient pressures,
such as those from long sampling lines. Methane required a large water vapor correction. Ethane could only be
reported for humidified samples, which affects the calibration protocol most often used in long-term operation. The
Aeris MIRA also had some software problems when not connected to the internet, so it requires regular attention.
However, overall the Aeris MIRA performed well when sampling plumes of incompletely combusted natural gas and
in large-scale ethane build-up overnight in the urban atmosphere. The small size and internal pump also make the
analyzer ideal for sampling from small mobile platforms such as cars and bikes (especially the Aeris MIRA LDS Pico,
which is the battery-powered version of the analyzer tested here).
While the Picarro G2210-i reported precise methane mole fractions and the analyzer performed adequately
in many of the tests, it could not detect ambient ethane enhancements of over 5 ppbv observed by the other instruments
in the polluted urban atmosphere. When sampling an incompletely combusted natural gas plume, it also reported a
reduction in ethane when the other analyzers reported a plume of ~10 ppbv.
Overall, we recommend the Aerodyne SuperDUAL or the Aeris MIRA Ultra LDS depending on the situation.
For long-term tower deployments or those with large mobile laboratories, the Aerodyne SuperDUAL provides the
best precision for methane and ethane. The other reported trace gases in the Aerodyne SuperDUAL, including CO,
carbon dioxide ($CO_2$) and nitrous oxide ($N_2O$) alongside ethane, also provide a way to more accurately attribute the
methane sources. For smaller mobile platforms, the Aeris MIRA is a more compact analyzer, and with careful use,
can quantify thermogenic methane sources to sufficient precision for short term deployments in urban or oil and gas
areas.
Data Availability. A permanent link will be added here once the permanent doi is available after the review process.
Currently the data from this study is available at: https://atmoscomp.ldeo.columbia.edu/content/data-sharing
Author Contributions. RC, AHD and LM designed the study, RC and AHD operated the instruments, AHD conducted
the tests, RC and AHD analyzed the data. RC prepared the manuscript with input from AHD and LM. The authors
declare that they have no conflict of interest.
Acknowledgements
Funding for this study was provided through contracts to the New York State Energy Research and Development
Authority (NYSERDA) grants #160536, #100413, #137484 and #183865, and National Oceanic and Atmospheric
Administration (NOAA) grant #NA20OAR4310306. We thank Ricardo Toledo-Crow and the Next Generation
Environmental Sciences Observatory of the Advanced Sciences Research Center, City University of New York for
Observatory space while conducting the instrument evaluations.

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
