# Peer review of "Intercomparison of commercial analyzers for atmospheric 1"

_Atmospheric Measurement Techniques, 2022_

## Author Comment (AC1)

**Response to RC1**

**General comments**
The paper report on tests and comparison of three recent optical analyzers dedicated to measuring both CH4 and C2H6. These sensors are increasingly needed for partitioning CH4 emissions from the fossil fuel sector. This is a very timely study since CH4 emissions from the oil and gas sector are under increasing scrutiny with the potential to achieve rapid climate change mitigation results.

The study has selected three available optical sensors that represent practical option now for field measurements and offer the possibility to run long term observations as well. The authors have brought careful consideration to properly running the instruments in their test environment. The tests offer insight into water vapour dependencies and corrections, precision, short term precision and stability against zero gas. The paper provide thoughtful considerations on practical aspects and compliance of the analyzers for different scientific purposes and contexts.
Overall the paper is well written, well presented and clear.
We thank the reviewer for all their helpful comments and suggestions. They have helped to make this a much better paper and we appreciate the time they spent on this paper.

I suggest to compare succinctly metrology definitions with those used in a large network with systematic pre-deployment sensor verification such as ICOS (e.g. Yver et al 2015 www.atmos-meas-tech.net/8/3867/2015/)
We thank the reviewer for this suggestion. While the ICOS definitions are useful for standard CO2/CH4 networks with well characterized instruments, they are not ideal for new instrument testing for selectivity. Most of our tests do not fit nicely into the ICOS metrology definitions. But Figure 2 where we evaluate precision (Allan-Werle Variance) is similar to continuous measurement repeatability (CMR), which is a repeatability measure applied to continuous measurements. Figure 3 where we evaluate nitrogen sampling over time is similar to short-term repeatability. We do not asses long-term repeatability, which would take months. In any case, we have added phrases to these sections to show the relation to each of the ICOS tests.

I have three main remarks about this study :
- In this study the instruments are not compared against a reference measurement technique such as GC for C2H6. This is a missed opportunity of the study to highlight the added value of these optical analyzers against GC dedicated to light alkanes such as ease of operation and time resolution. On the other hand comparing the results of each sensor against GC performance would have been extremely instructive. Without this comparison it is less conclusive whether the best optical sensors are more adapted than GC for long term observations of background atmospheric composition or analysis of airmasses of regional representativity.
Unfortunately, we did not have access to a GC based instrument for ethane when conducting this intercomparison study. GCs are difficult to use on mobile platforms and are not widely used in the US so to be honest, we never even thought about trying to

procure one. The GC-FIDs measuring ambient ethane in New York are designed for hourly monitoring of ethane and do not measure methane. These instruments are located in state equivalent EPA monitoring sites with little space or restricted access, which makes it difficult to conduct testing and intercomparisons with those instruments. We will look into this comparison for future studies.

- The paper lacks discussion on linearity especially toward large concentrations that could be useful in the context of industrial applications. Do the three sensors perform equally linearly within the full concentration ranges that can be observed in near-source studies for both species?

We thank the reviewer for this suggestion. It was not possible to calibrate the methane and ethane to the high mixing ratios observed in these plumes as they are outside the range of our calibration standards. Instead, we have compared the Aeris and Aerodyne instruments for ambient plume sampling in Feb 20-21 at 1s for methane and 10s for ethane and the relationship is linear (Fig S9). This result is representative for the ambient sampling periods (including much more data caused the 1s data plots to be huge). While this does not directly test linearity, it is unlikely that the instruments would have identical non-linearities and therefore strong correlations between reported concentrations from the instruments likely indicates that they retain linear behavior well beyond the range of the span. We have added a discussion of the linearity of the Aerodyne and Aeris instruments at the end of Section 3.2 Instrument calibration.

"We evaluated the linearity of the instruments outside our range of calibration standards by comparing the instrument response for the Aerodyne SuperDUAL and Aeris MIRA during the high plumes (as discussed in Section 3.5 below). Fig S3 shows the linearity of 1s methane and 10s ethane for Feb 20-21, 2022 with the Aeris MIRA and Aerodyne SuperDUAL. The methane fit (Slope 1.002) is slightly closer to 1 than the ethane fit (Slope: 1.048 +/- 0.002). The slow response of the Picarro G2210-i meant that it could not represent plumes of ethane at sufficient resolution to allow for valid comparison. While this does not directly test linearity, the strong correlations between reported concentrations from the instruments likely indicates that they retain linear behavior well beyond the range of our calibration standards. "

[Figure]

Figure S3: Linearity of (a) methane (1s data) and (b) ethane (10s aggregated data) for two days in February 20-21, 2022. The 1:1 line is shown in black. The slope and intercept calculated from an ordinary least squares with 95% confidence intervals are shown in the top left. The calculated slope and intercept are shown as a red dashed line.

- The obvious problem about the Picarro not detecting the ambient air 10ppb C2 peaks is a major concern (section 3.5). Is it representative of a shortcoming of the G2210 model, or is it just a problem with this particular unit at that particular time, mishandling

or poor quality control at the factory by the manufacturer? This is an important concern that is opened by this study but is frustratingly not really addressed. Such a poor performance is useful to publish since this particular case suggest that the instrument do not work according to expectations. However if this is behaviour is not representative of the G2210 model it cannot left as it is. Further work is needed that investigates whether this poor performance is to be attributed either to shortcomings of the G2210 model or to another reason (and if there is another reason, is there hope that it can be corrected by the manufacturer or the user). In my opinion, before this paper is published this work should really be completed to address this point. Ideally by running in parallel another G2210 unit that could maybe be borrowed from the manufacturer or elsewhere. This could be done even through a few days worth of measurements close to a known C1/C2 emitter (natural gas industrial site) with only the 2 picarros.

We thank the reviewer for this comment. We struggled with how to represent the behavior of the Picarro fairly and may have erred on the side of not including enough information. We have no reason to believe that the behavior described here is specific to this analyzer. In the few papers that reported using the G2210-i analyzer (e.g. Lebel et al., 2022, Defratyka et al., 2021), no ethane data has ever been plotted/shown in a figure. Methane reported by the analyzer is within spec provided by Picarro and the methane isotopes (not evaluated here) also seem to be reporting accurately (based on some brief testing at the isotope lab in Rochester).

This particular G2210i analyzer was operating for over a year at a background site where it was reporting somewhat unexpected behavior (negative ethane concentrations, anticorrelations of ethane with methane, etc). The instrument was returned to Picarro for service, where it was kept for a few months before it was returned to the PI and we conducted this study soon after. When we received the analyzer, we were assured by Picarro engineers that it was functioning as expected and completely within specifications. What we don't show here is all the negative ethane concentrations and the anticorrelation with methane (obviously a malfunction) that we observed in the first two weeks of our study. We contacted Picarro after the two weeks and spent over ten hours on various meetings with engineers and scientists at Picarro trying to understand the behavior described here, specifically showing them the negative response for ethane vs the other analyzers. We informed them that we were working on this manuscript and gave them opportunity to deal with the problem before we submitted. However, they failed to identify a problem, other than it *might* be a CO or VOC interference from the combustion, and no solution was presented. They never offered to send a second analyzer for additional comparison and we would not suggest that anyone should spend money buying one. The analyzer was returned to the background site (with little CO) after this study and seems to be operating within specifications since then.

We have added the following (blue) text to Section 3.5:

"In order to test the suitability of each analyzer to report accurate methane and ethane mole fractions in ambient air, we ran all instruments sampling ambient air from the CUNY Observatory in Harlem, NY, for 3-4 weeks in February 2022. In general, air is cold and very dry in New York City in winter and it took some time to learn that we had to humidify the Aeris MIRA and Picarro G2210-i sample flows in order to record valid data (see instrument characterization experiments described above). The Picarro G2210-i was oftenreporting negative ethane and

negative correlations of ethane with methane for the first two weeks of observations. We then requested that Picarro engineers check the instrument and they assured us it was performing as expected. So we have focused on Feb 17-22, 2022 (see Fig S8), when the G2210-i was confirmed to be performing to specification. Figure 4 shows typical zoomed in examples of the ambient methane and ethane mole fractions observed by all the analyzers when sampling ambient air in February 2022."

[Figure]

Fig S8: Time series of ambient sampling of all three instruments for Feb 17 – 21, 2022 (Time in UTC). (a) Methane ($CH_4$, ppbv), (b) ethane ($C_2H_6$, ppbv) and (c) ethane/methane ratio ($C_2H_6/CH_4$, %). Compressed air tanks were sampled on Thursday and Monday. The Aerodyne SuperDUAL data (black line) is shown at 1s, the Aeris MIRA (blue circle) is a 10s average and the Picarro G2210-i is 1s for (a) $CH_4$ and (b) 5 minute average for $C_2H_6$. All three analyzers observed plumes of methane on Friday night (Feb 18[th]) into Saturday morning (Feb 19[th]). While the Aerodyne SuperDUAL and Aeris MIRA also saw increases in ethane that identified these plumes as natural gas, the Picarro G2210-I did not. The Picarro G2210-i also reported a decrease in ethane when sampling a compressed air cylinder, contrary to the increase reported by the Aerodyne SuperDUAL and Aeris MIRA.

We have no opinion on how the G2210-i will do at natural gas industrial sites. That is not the topic of this study and we disagree with the Reviewers suggestion that two G2210-i analyzers should be tested at a natural gas source. We are specifically discussing here that these analyzers are not suitable for *urban* environments or areas with natural gas *combustion*, both of which have large possible interferents for the ethane reported by the analyzer. It is important that any analyzer for use in an urban or combustion environment would be evaluated in that specific environment. We edited a line in Section 3.4 to emphasize this point.

"Our results indicate that the Picarro G2210-i should not be used to selectively measure ethane near combustion sources such as flares, or natural gas power plants or in urban areas that combust natural gas on a large scale. Indeed, care should be taken to ensure that thermogenic sources are not erroneously attributed to biogenic sources with the Picarro G2210-i in urban areas."

**Specific comments**

Abstract - I would encourage the authors to provide numeric values for precisions in the abstract

We made a conscious decision not to list precisions in the abstract. All instrument precisions are in the sub-ppb range for both methane and ethane but the Picarro isn't selectively measuring ethane in ambient air, which is the most important metric. Listing precisions could give a false sense that the Picarro actually works (which it doesn't). We can list the precisions if the editor requires it but we would prefer not to for the reasons stated above.

L44 the statement could reflect the nuance that biomass burning co emits CH4 and ethane

Good point. We have edited the text as follows:

"The incomplete combustion of liquid (e.g. natural gas) or solid (e.g. coal, wood) fuels can co-emit high concentrations of carbon monoxide (CO) and other Volatile Organic Compounds VOCs."

L81 please explain the values of 2 and 1.45

The water broadening coefficient is a correction applied to the fitting of the absorption spectra that is a function of the water vapor observed in the cell. The coefficient is experimentally determined by the manufacturer and applied to generate the dry mole fraction reported methane and ethane. A similar correction is used in all laser-based spectrometers. We have edited the text as follows:

"We use the default water broadening coefficient (WBC) for all species (WBC = 2), except CO (WBC = 1.45)."

L81 define footprint in this context

We meant the table top area needed for the instrument but we have edited the text to avoid confusion:

"The analyzer is large and heavy"

L95 typically could an external ush pump solve the problem?

Yes! And I believe a group at Scripps are currently working with Aeris to develop a "tower ready" version of the instrument with an external pump to avoid this problem in future. But it's not possible to do right now because of interactions with the internal pump.

L114 should be lowest humidity or driest air,

Good catch. Corrected to lowest humidity.

L243 it should be noted here that the G2201-i, unlike the G2210 is not intended for ethane measurements. Is the value reported for Defratyka et al. applying any specific correction? Are the two sensors similarly configured in terms of laser wavelengths?

Despite asking, we never did get the ethane wavelength for the G2210-i so we can't answer this question. After re-reading the Defratyka et al paper, we noticed that we mis-quoted the precision, which should have been 12 ppbv (1 minute). We have removed the following sentence as the comparison is not helpful and, as the reviewer states, the G2201-i was not designed to measure ethane (even if Defratyka et al do an amazing job quantifying the ethane response).

**Section 3.4 Would measuring a gas cylinder with concentration typical of ambient air yield similar or different conclusion about long term stability?**

We have added more discussion to Section 3.4 to include discussion of the comparison with the compressed gas cylinder experiments (Fig 1, 2, S4- S7). The Aeris results are consistent for both the air and nitrogen for ethane but not methane. The Picarro results are different for methane and at the suggestion of another reviewer, we have included a new paragraph discussing those results.

"Using a Picarro G1301, (Nara et al., 2012) observed a pressure broadening effect when sampling gas with a range of oxygen and argon that resulted in a ~2 ppb bias in methane. We would expect to see a larger pressure broadening effect when sampling dry nitrogen free of oxygen and argon, which may explain some of the variability in Fig 3a. Indeed, there is no increased variability in methane observed by the Picarro G2210-i when sampling from a compressed air cylinder at low humidity (Fig 1(a)). For the Aeris MIRA we see different behavior for the methane and ethane. The ethane results are consistent for both compressed air and nitrogen with more ethane variability at low humidity. The methane variability is much larger when sampling humidified nitrogen and dry compressed air than seen when sampling dry nitrogen and humidified compressed air (see Fig 1 and S1). In our tests here, the G2210-i stability for methane is the best of the three analyzers when sampling humidified nitrogen boil off, which indicates that the addition of nitrogen from a dewar is possible as a long-term zero only if the flow is humidified. However, for the Aeris MIRA, we observe much more methane variability in humidified nitrogen and lots of ethane variability in dry nitrogen so we do not recommend using nitrogen as a long-term zero."

**L287 : difficult to see the longer CH4 peak duration in the figure**

We have added Figure S9 to the supplement with a zoomed in 20 mins around the peak of this plume. Fig S9 shows the slower reduction in methane from the Picarro vs the other instruments. At the request of another reviewer, we also included water vapor and CO to the plot.

[Figure]

"Figure S9: Time series of ambient sampling of all three instruments for 13:10 – 13: 30 UTC Feb 18, 2022. (a) Methane ($CH_4$, ppbv) and water vapor (cyan, % measured on SuperDUAL, 1Hz), (b) ethane ($C_2H_6$, ppbv) and carbon monoxide (CO, ppbv; measured on SuperDUAL, 1Hz). The Aerodyne SuperDUAL data (black line) is shown at 1s, the Aeris MIRA (blue circle) is 1s average and the Picarro G2210-i is (a) 1s, (b) 10s and 5 minute average. This is a zoomed in version of Figure 4 (a) and (b). The Picarro G2210-i $CH_4$ is slow to return to background concentrations due to the low flow rate. The $C_2H_6$ reported by the Picarro G2210-i is reduced during the increase observed by the Aeris MIRA and Aerodyne SuperDUAL instruments plume."

L289 This is noticeably incompatible with the 0.8ppb precision reported in previous sections

Agreed. This result suggests that precision alone is insufficient to determine instrument viability for a given environment. It is important to also determine *selectivity* in order to understand instrument performance.

L304: For the picarro, not being applicable close to sources is a strong problem when considered the 0.8ppb C2H6 precision: such a precision orients the applicability of this analyzer to near-source measurements

As stated above, we are specifically discussing here that these analyzers are not suitable for *urban* environments or areas with natural gas *combustion*, both of which have large possible interferents for the ethane reported by the analyzer. It is important that any analyzer for use in an urban or combustion environment would be evaluated in that specific environment. As stated above we have edited this line to emphasize this point.

"Our results indicate that the Picarro G2210-i should not be used to selectively measure ethane near combustion sources such as flares, or natural gas power plants or in urban areas that combust natural gas on a large scale. Indeed, care should be taken to ensure that thermogenic sources are not erroneously attributed to biogenic sources with the Picarro G2210-i in urban areas."

L328, continuing on my general comments: is this problem specific to a deficient unit or representative of all G2210? as it is, it cannot be concluded whether the picarro technique is performing poorly or if this particular unit has a problem. Has it been check with the manufacturer? It would be useful to reproduce test with another unit.

We believe this behavior described here is not just for this analyzer. Please see the discussion above. It has been checked by the manufacturer and I disagree with suggesting someone else should buy one a G2210-i to measure ethane in an urban environment.

**Editorial**
Table 1 Aeris CH4 formula: would there be a way to better present the different formulas: linear vs quadratic Fig 2 x axis would be more useful labelled in seconds.

We have changed the x axis label.
We struggled with how best to represent the different formulas in Table 1 and this was the best we could think of due to the non-linearity of the Aeris methane. If anyone has a better suggestion, we'd be happy to change the table.

L242 please choose unit ppb or nmol mol-1
Done. Sorry about that.

Fig 4 there seems to be a technical problem with the figure: it might be not well constructed.

It was fine when we uploaded and printed it but we will work with the editors to ensure it is clear before publication.

L327 TestS plural
Done.

---

## Author Comment (AC2)

**Response to RC2:**

Commane et al present a study comparing three commercially available analyzers for atmospheric CH4 and C2H6. Results from typical laboratory performance tests and calibrations are discussed. The authors also present results from a very short period of real-world data collected in an urban area.

The paper is well-written and clearly structured. Core results are highlighted and the logic is easy to follow. The description of the experiments is good, but could be improved slightly with some additional details. Overall, this a very good example of a study that other experimental scientists in this field can use to choose suitable equipment for future work. Testing analyzers with the combination of gases CH4 and C2H6 is very timely as there is an increasing amount of research on local atmospheric methane signals in recent years and many these studies could be improved if reliable source apportionment (thermogenic vs microbial) was done. Thus, this paper is highly suitable for AMT after minor comments are addressed.

We thank the reviewer for their helpful comments and suggestions that have improved the manuscript.

General comments:

Please consider using consistent unit for the mole fractions reported here. Sometimes it is reported as ppbv, then as ppb in the figures and nmol mol-1 in some tables. Also please add an explanation that nmol mol-1 is only equivalent to ppbv assuming ideal gases (which C2H6 and CH4 are not) or consistently report in nmol mol-1.

Sorry about this confusion. We have made the reported units consistent across the manuscript. As the reviewer mentions, the instruments measure dry mole fractions (nmol mol$^{-1}$) but as the analyzers report mixing ratios and these units are most commonly used in the field, we have chosen to report all units as ppbv. We have added a sentence to clarify in Section 2.1

"Each of the analyzer described below reports the dry mole fraction of methane and ethane in air using units of ppbv, parts-per-billion by volume, which is the equivalent of nmol mol$^{-1}$ for an ideal gas."

Please specify the composition of the zero air at least once to ensure that it is not just a synthetic N2/O2 mixture but has the correct (ambient air) amount of all three matrix gases.

Thanks for catching this. The Zero tests were done with nitrogen and not zero air. To avoid confusion we have renamed Section 2.2.4 Zero-air tests to

2.2.4 Nitrogen tests

Line 245 was changed to:

"We evaluated the stability of frequent additions of nitrogen (liquid nitrogen boil-off free of methane, ethane, $CO_2$, etc.) for all three analyzers."

Please add to the discussion about the Picarro performance that the tests using N2 should be disregarded as their CRDS systems have been shown to be susceptible to changes in matrix gas, see Nara et al. 2012 https://doi.org/10.5194/amt-5-2689-2012 We should not expect for the CRDS system to work properly for any mixture with missing O2 or Ar.

We thank the reviewer for highlighting this point. Nara et al., 2012 calculate a +/-2 ppbv decrease in reported $CH_4$ due to pressure broadening effects when changing O2 and Ar from the sampled gas of a G1301 analyzer. We're not sure the same correction is observed for the wavelengths used for the detection of methane here. If it did, we should see more bias for the reported methane in Fig 3. But the Picarro does particularly well in the humidified nitrogen. We clarified in Section 2.2.4 that there may be trace levels of O2 and Ar in the LN2 (see the helpful reviewer comment below). We also edited the text at the end of the second (Picarro) paragraph in Section 3.4.

"Using a Picarro G1301, (Nara et al., 2012) observed a pressure broadening effect when sampling gas with a range of oxygen and argon that resulted in a ~2 ppb bias in methane. We would expect to see a larger pressure broadening effect when sampling dry nitrogen free of oxygen and argon, which may explain some of the variability in Fig 3a. Indeed, there is no increased variability in methane observed by the Picarro G2210-i when sampling from a compressed air cylinder at low humidity (Fig 1(a)). For the Aeris MIRA we see different behavior for the methane and ethane. The ethane results are consistent for both compressed air and nitrogen with more ethane variability at low humidity. The methane variability is much larger when sampling humidified nitrogen and dry compressed air than seen when sampling dry nitrogen and humidified compressed air (see Fig 1 and S1). In our tests here, the G2210-i stability for methane is the best of the three analyzers when sampling humidified nitrogen boil off, which indicates that the addition of nitrogen from a dewar is possible as a long-term zero only if the flow is humidified. However, for the Aeris MIRA, we observe much more methane variability in humidified nitrogen and lots of ethane variability in dry nitrogen so we do not recommend using nitrogen as a long-term zero."

Specific comments:

Line 134: How pure is the N2 boil off? could there be other contaminant gases than CO?

It is possible that there are species other than CO (boiling point (b. p.) is -191.5°C) in the LN2 boil off (b. p. $N_2$ -196°C) such as argon (-186°C) or oxygen (b. p. -183°C) but we have

not measured any other species with the instruments we have used at the site. We have clarified Section 2.2.4 with the following text:

We use the boil off from a large liquid nitrogen dewar, which can be refilled on site, and which contains a variable mole fraction of carbon monoxide (~250 ppbv), and may contain trace levels of oxygen and argon.

Line 135: The units for carbon monoxide should be nmol mol-1 or ppb

Agreed. We have used ppbv throughout the manuscript for the reasons explained above.

Line 173: See general comments on use of different units throughout the manuscript

Agreed. We have cleaned up the use of units throughout the manuscript.

Line 176f: different units in table, figure and caption seems unnecessary

Agreed. We have cleaned up the use of units throughout the manuscript.

Line 213: Consider adding information about the actual measurement cycle of the Picarro. The data rates does not automatically match the measurement frequency here.

We have added the following text to Section 2.1.3

"The measurement and reporting cycle of the Picarro G2210-i are 1Hz. But the low flow rate reduces the instrument response time considerably. We have corrected for the delay and report methane at 1Hz and we have averaged the ethane to 10s and 5 minutes."

Line 249 - Figure3: See comments on using CRDS analyzers for a non-natural air matrix

Agreed. Edited text is mentioned above.

Line 273 - Figure4: As water vapour has been shown to be a critical component please add the H2O levels to the plot. Are there any changes in H2O reported by the Picarro during the expected C2H6 peaks. Previous studies have shown strong dependencies. https://doi.org/10.5194/amt-14-5049-2021

We have corrected each instrument for water vapor dependence (as per Section 3.1). But, unlike Fig 3 in Defratyka et al., 2021, the water vapor dependence of the ethane reported by the G2210-i was negligible (See Figure 1) so we do not expect a water vapor response in the ethane signal. Adding water to Fig 4 was a bit messy so we have shown the reported water in the zoomed in version (Fig S9). We also added CO to highlight the combustion character of the plume.

[Figure]

"Figure S9: Time series of ambient sampling of all three instruments for 13:10 – 13: 30 UTC Feb 18, 2022. (a) Methane ($CH_4$, ppbv) and water vapor (cyan, % measured on SuperDUAL, 1Hz), (b) ethane ($C_2H_6$, ppbv) and carbon monoxide (CO, ppbv; measured on SuperDUAL, 1Hz). The Aerodyne SuperDUAL data (black line) is shown at 1s, the Aeris MIRA (blue circle) is 1s average and the Picarro G2210-i is (a) 1s, (b) 10s and 5 minute average. This is a zoomed in version of Figure 4 (a) and (b). The Picarro G2210-i $CH_4$ is slow to return to background concentrations due to the low flow rate. The $C_2H_6$ reported by the Picarro G2210-i is reduced during the increase observed by the Aeris MIRA and Aerodyne SuperDUAL instruments plume."

Line 303: Generally restricting the use of G2201-i Picarros in certain regions seems an extreme suggestion given that the real-world test period was extremely short and only performed in one region.

We do not comment on the G2201-i as it was not evaluated here. We are asserting that the G2210-i evaluated here is not appropriate for use in some conditions such as the specific urban polluted air as sampled here. We have added a new Figure S8 to show some of the longer time series of observations when all three instruments were sampling ambient air and we have added the following (blue) text to Section 3.5:

"In order to test the suitability of each analyzer to report accurate methane and ethane mole fractions in ambient air, we ran all instruments sampling ambient air from the CUNY Observatory in Harlem, NY, for 3-4 weeks in February 2022. In general, air is cold and very dry in New York City in winter and it took some time to learn that we had to humidify the Aeris MIRA and Picarro G2210-i sample flows in order to record valid data (see instrument characterization experiments described above). The Picarro G2210-i was oftenreporting negative ethane and negative correlations of ethane with methane for the first two weeks of observations. We then requested that Picarro engineers check the instrument and they assured us it was performing as expected. So we have focused on Feb 17-22, 2022 (see Fig S8), when the G2210-i was confirmed to be performing to specification. Figure 4 shows typical zoomed in examples of the ambient methane and ethane mole fractions observed by all the analyzers when sampling ambient air in February 2022."

[Figure]

Fig S8: Time series of ambient sampling of all three instruments for Feb 17 – 21, 2022 (Time in UTC). (a) Methane ($CH_4$, ppbv), (b) ethane ($C_2H_6$, ppbv) and (c) ethane/methane ratio ($C_2H_6/CH_4$, %). Compressed air tanks were sampled on Thursday and Monday. The Aerodyne SuperDUAL data (black line) is shown at 1s, the Aeris MIRA (blue circle) is a 10s average and the Picarro G2210-i is 1s for (a) $CH_4$ and (b) 5 minute average for $C_2H_6$. All three analyzers observed plumes of methane on Friday night (Feb 18[th]) into Saturday morning (Feb 19[th]). While the Aerodyne SuperDUAL and Aeris MIRA also saw increases in ethane that identified these plumes as natural gas, the Picarro G2210-I did not. The Picarro G2210-i also reported a decrease in ethane when sampling a compressed air cylinder, contrary to the increase reported by the Aerodyne SuperDUAL and Aeris MIRA.

As stated for Reviewer 1:

We struggled with how to represent the behavior of the Picarro fairly and may have erred on the side of not including enough information. We have no reason to believe that the behavior described here is specific to this analyzer.
In the few papers that reported using the G2210-i analyzer (e.g. Lebel et al., 2022), no ethane data has every been plotted/shown in a figure. Methane reported by the analyzer is within spec provided by Picarro and the methane isotopes (not evaluated here) also seem to be reporting accurately (based on some brief testing at the isotope lab in Rochester).
        This particular G2210i analyzer was operating for over a year at a background site where it was reporting somewhat unexpected behavior (negative ethane concentrations, anticorrelations of ethane with methane, etc). The instrument was returned to Picarro for service, where it was kept for a few months before it was returned and we conducted this study soon after. When we received the analyzer, we were assured by Picarro engineers that it was functioning as expected and completely within specifications. What we don't show here is all the negative ethane concentrations and the anticorrelation with methane (obviously a malfunction) that we observed in the first two weeks of our study. We contacted Picarro after the two weeks and spent over ten hours on various meetings with engineers and scientists at Picarro trying to understand the behavior described here, specifically showing them the negative response for ethane vs the other analyzers. We informed them that we were working on this manuscript and gave them opportunity to deal with the problem before we submitted. However, they failed to identify a problem,

other than it *might* be a CO or VOC interference from the combustion, and no solution was presented. They never offered to send a second analyzer for additional comparison and we would not suggest that anyone should spend money buying one. The analyzer was returned to the background site (with little CO) after this study and seems to be operating within specifications since then.

Line 313: Please provide the citation for the study that established the performance equivalency of the Aerodyne and Aerodyne 'mini' system.

We have added a link to the Aerodyne spec sheet for the mini that shows the precision and edited the text as follows:

There is a smaller size instrument from Aerodyne – the Aerodyne "mini" – which has a methane/ethane precision between that of the SuperDUAL and the Aeris MIRA but this also requires an external chiller and large pump (see https://www.aerodyne.com/wp-content/uploads/2021/11/Ethane.pdf; 60s precision of 0.05 ppbv $CH_4$ and 0.015 ppbv $C_2H_6$).

---

## Author Comment (AC3)

**Response to CC1: Manuele Polli**

I enjoyed reading this work that address the need for accurate methane and ethane analyzers to monitor natural and anthropogenic green house gas emissions. Commane R. et al. assess this need by comparing three different commercial analyzers – Aerodyne SuperDUAL, Aeris Mira Ultra LDS, Picarro G2210-i. The authors tested the water sensitivity, instrument calibration, long-term instrument stability and accuracy in ambient sampling for each instrument. The results are convincing and the methodology adopted were well described. The clear language as well as the use of appropriate figures are strengths of this paper. I suggest the authors to put more effort into the description of the need of this study as well as the conclusion following the achieved results. Based on those suggestions and the following comments I recommend that this paper by Commane R. et al. to be accepted after minor revision.

Major comments

1. Good informative title!

It may be a bit boring but it seems to work!

2. Very good, short and precise Abstract containing all the necessary information!
3. Introduction: Easy to follow even if one is not from the field, great wording! Main concern of the introduction is the necessity of this study. What research for instruments have been done before i.e. back the necessity of this study with other sources like doing in the beginning of the methods section (e.g. line 69-71 & for Picarro – line 102). Or also adding why these new commercial analyzers are better then the previous ones and the need of a intercomparison compared to previous studies.

We thank the reviewer for this suggestion. We have added some text to the introduction to highlight the relevance of this study:
As far as we can tell, there has not yet been a systematic assessment and characterization of these newly available laser-based ethane spectrometers. There is also little guidance available to those now charged with instrumenting networks and mobile platforms for methane source apportionment.

4. Good methodology, following a good structure by firstly giving technical information, then the ideal use, possible complications followed by a detailed approach to your analysis.

Thanks

5. The structure and visualization (especially Figure 4) of your results are great. The section "3.4 Long-term instrument stability" misses a discussion and is only describing the results present. Otherwise your descriptions and discussions were short & precise, well done!

Based on some of the other review comments we have added a short discussion to Section 3.4 for the Picarro performance in particular. We have also added a short discussion about the overall instrument performance.

"Using a Picarro G1301, (Nara et al., 2012) observed a pressure broadening effect when sampling gas with a range of oxygen and argon that resulted in a ~2 ppb bias in methane. We would expect to see a larger pressure broadening effect when sampling dry nitrogen free of oxygen and argon, which may explain some of the variability in Fig 3a. Indeed, there is no increased variability in methane observed by the Picarro G2210-i when sampling from a compressed air cylinder at low humidity (Fig 1(a)). For the Aeris MIRA we see different behavior for the methane and ethane. The ethane results are consistent for both compressed air and nitrogen with more ethane variability at low humidity. The methane variability is much larger when sampling humidified nitrogen and dry compressed air than seen when sampling dry nitrogen and humidified compressed air (see Fig 1 and S1). In our tests here, the G2210-i stability for methane is the best of the three analyzers when sampling humidified nitrogen boil off, which indicates that the addition of nitrogen from a dewar is possible as a long-term zero only if the flow is humidified. However, for the Aeris MIRA, we observe much more methane variability in humidified nitrogen and lots of ethane variability in dry nitrogen so we do not recommend using nitrogen as a long-term zero."

6. Conclusions and Recommendations: Brief and containing all the major findings, very good!

Thanks!

Main concern – You do not recommend the usage of Picarro compared to both other instruments but only based on the 2 unrealistic ethane measurements during the plumes. If I understand correctly all the other tests and calibrations were qualitatively indifferent or not significantly worse when using Picarro. Following those findings, I would conclude that future research on the Picarro instrument measuring ethane plumes is needed (as you "only" measured twice) rather than only recommending the other two instruments. One idea could also be to leave the ambient sampling experiment out of the paper until more samples and a concluding review can be reached.

We thank the reviewer for highlighting this point. We had struggled with how to best describe the behavior of the G2210-i fairly and now realize that we didn't include enough information. We had 4 weeks of ambient sampling in February 2022 but for the first two weeks, the G2210-i instrument either reported negative ethane concentrations or showed anticorrelation of ethane with methane (when the Aeris and Aerodyne showed positive correlations). The two examples shown in Figures 4 are just short examples of the longer-term ambient sampling, zoomed in for ease of viewing. We saw many of these type of events in Feb 2022. We have now added Fig S8 that shows 5 days of all three instruments sampling ambient air in NYC. The same G2210-i was also operated at an air quality site for over a year and showed similar behavior (negative ethane during plumes).

[Figure]

Fig S8: Time series of ambient sampling of all three instruments for Feb 17 – 21, 2022 (Time in UTC). (a) Methane (CH4, ppbv), (b) ethane (C2H6, ppbv) and (c) ethane/methane ratio (C2H6/CH4, %). Compressed air tanks were sampled on Thursday and Monday. The Aerodyne SuperDUAL data (black line) is shown at 1s, the Aeris MIRA (blue circle) is a 10s average and the Picarro G2210-i is 1s for (a) CH4 and (b) 5 minute average for C2H6. All three analyzers observed plumes of methane on Friday night (Feb 18th) into Saturday morning (Feb 19th). While the Aerodyne SuperDUAL and Aeris MIRA also saw increases in ethane that identified these plumes as natural gas, the Picarro G2210-I did not. The Picarro G2210-i also reported a decrease in ethane when sampling a compressed air cylinder, contrary to the increase reported by the Aerodyne SuperDUAL and Aeris MIRA.

As stated in the response to Reviewer 1:

We struggled with how to represent the behavior of the Picarro fairly and may have erred on the side of not including enough information. We have no reason to believe that the behavior described here is specific to this analyzer. In the few papers that reported using the G2210-i analyzer (e.g. Lebel et al., 2022), no ethane data has ever been plotted/shown in a figure. Methane reported by the analyzer is within spec provided by Picarro and the methane isotopes (not evaluated here) also seem to be reporting accurately (based on some brief testing at the isotope lab in Rochester).

This particular G2210i analyzer was operating for over a year at a background site where it was reporting somewhat unexpected behavior (negative ethane concentrations, anticorrelations of ethane with methane, etc). The instrument was returned to Picarro for service, where it was kept for a few months before it was returned and we conducted this study soon after. When we received the analyzer, we were assured by Picarro engineers that it was functioning as expected and completely within specifications. What we don't show here is all the negative ethane concentrations and the anticorrelation with methane (obviously a malfunction) that we observed in the first two weeks of our study. We contacted Picarro after the two weeks and spent over ten hours on various meetings with engineers and scientists at Picarro trying to understand the behavior described here, specifically showing them the negative response for ethane vs the other analyzers. We informed them that we were working on this manuscript and gave them opportunity to

deal with the problem before we submitted. However, they failed to identify a problem, other than it *might* be a CO or VOC interference from the combustion, and no solution was presented. They never offered to send a second analyzer for additional comparison and we would not suggest that anyone should spend money buying one. The analyzer was returned to the background site (with little CO) after this study and seems to be operating within specifications since then.

We have added the following (blue) text to Section 3.5:

"In order to test the suitability of each analyzer to report accurate methane and ethane mole fractions in ambient air, we ran all instruments sampling ambient air from the CUNY Observatory in Harlem, NY, for 3-4 weeks in February 2022. In general, air is cold and very dry in New York City in winter and it took some time to learn that we had to humidify the Aeris MIRA and Picarro G2210-i sample flows in order to record valid data (see instrument characterization experiments described above). The Picarro G2210-i was oftenreporting negative ethane and negative correlations of ethane with methane for the first two weeks of observations. We then requested that Picarro engineers check the instrument and they assured us it was performing as expected. So we have focused on Feb 17-22, 2022 (see Fig S8), when the G2210-i was confirmed to be performing to specification. Figure 4 shows typical zoomed in examples of the ambient methane and ethane mole fractions observed by all the analyzers when sampling ambient air in February 2022."

{minor comments}

General comment: You're talking about concentrations of methane and ethane, so I suggest to add that in the text e.g. Line 57: "Here, we evaluated three laser- based spectrometers that are marketed to measure ambient ethane and methane "concentration".

Technically the instruments report dry mole fractions of those gases (ppbv) and not concentration (molecule cm$^{-3}$) so we have added the text:

"Here, we evaluated three laser-based spectrometers that are marketed to measure ambient dry mole fractions of ethane and methane.

General comment: Be consistent using either ppbv or ppb (e.g. Table 1 & Figure 1 contain ppb but description and text contains ppbv)

Done. Sorry about that confusion.

General comment: Keep the capitalization within your titles consistent (e.g. Line 163 "Characterization of water sensitivity" or Line 126 "Instrument stability")

Corrected.

Abstract: Good and compact but some sentences (e.g. starting Line: 13, 16, 21) are very lengthy which could be rephrased into two sentences.

We shortened a number of sentences in the abstract.

"The Aerodyne SuperDUAL instrument performed best of the three instruments but it is large and requires expertise to operate… The more compact Aeris MIRA can, with careful use, quantify thermogenic methane sources to sufficient precision for mobile or short term deployments in urban or oil and gas areas. We weighed the advantages of each instrument, including size, power requirement, ease of use on mobile platforms, and expertise needed to operate the instrument. We recommend the Aerodyne SuperDUAL or the Aeris MIRA Ultra LDS depending on the situation."

Line 9 – No point after the title Abstract

Done.

Line 12 – different sources not difference

Done.

Introduction: Explain the term nocturnal boundary layer as it gets mentioned a few times in the result & discussion section.

We reworded those sentences in the Discussion to make it less jargony. Instead we used terms like "build up in the urban atmosphere overnight".

Line 46 – add sources "to many studies"

Done.

Line 50 – no "e.g." needed

Done.

Line 52 – source?

Added a list of sources for Boston, DC, LA, and Indianapolis.

Methods: Line 71 – sentence is very lengthy.

We have split the sentence in two:

These spectrometers use a continuous wave interband cascade laser (ICL) based spectrometer to measure methane, ethane and water vapor. They are often used in a two laser system alongside a continuous wave quantum cascade laser (QCL) to measure dry mole fractions of carbon dioxide ($CO_2$), carbon monoxide (CO), and nitrous oxide ($N_2O$).

Line 90 – "There are..." what should this sentence contribute to the understanding of Aeris technology?

The technical specification of the Pico is very similar to the Ultra. So we have added the phrase:

There are few descriptions of the Aeris MIRA but Travis et al., 2020 described a similar, portable version of the instrument with an onboard battery (MIRA Pico, not evaluated here).

Line 111 – sentence is very lengthy.

We have split the sentence:

A counter flow of air was drawn through the Nafion at ~2000 sccm using a vacuum pump. The inlet to the counter flow was alternatively sampling the top of a container of water that was at a temperature slightly warmer than the observatory or dry air-conditioned air in the observatory.

Line 120 – Reword "calibrated" as it appears 2 times in the same sentence.

Agreed. We have reworded the sentence:

"Each of the instruments sampled two ambient range cylinders calibrated by…"

Line 129 – why was the regular zero not performed? – give reasoning

We stopped the zeros to allow or a direct comparison of all analyzers. If the laboratory temperature is stable (as on this day), a 4 hour time period without zeros is not detrimental to the performance of the SuperDUAL as seen in Fig 2. We have edited the text to:

"During this time the regular zero for the Aerodyne SuperDUAL was not performed to allow for direct comparison of all instruments."

Line 136 – Rewrite that sentence for clarity, i.e. I don't understand the meaning behind it.

We have rephrased the sentence:

"Regular nitrogen sampling is not required for the long-term operation of either the Picarro G2210-i or the Aeris MIRA. We evaluated the performance of the Aeris MIRA and Picarro G2210-i when sampling dry and humidified nitrogen."

Results and Discussion: Well written first paragraph (Line 160)!

Thank you!

Line 178 – Figure caption misses a point in the end.

Done.

Line 182 – Missing indent

Done.

Line 186 – Do you have a source to the possibility of laser wavelength drift?

We have seen similar behavior with the SuperDUAL when the reference lock is disabled. So it's a possibility but without a laser wavelength monitor (which we do not have), I don't know how we would confirm that theory. We discussed the problem in depth with engineers at Aeris Technologies and they have developed a fix for this problem by changing the lock to either the stronger water line (often saturated in normal operation) or locking to the methane. We have updated the text as follows:

"After discussion with engineers at Aeris Technologies, we learned that there are two water vapor peaks in the spectral window. So this problem could be mitigated when sampling dry cylinders by locking to the stronger water vapor absorption peak, which is often saturated during normal operation, or to the methane line directly. Note that locking to the methane line would prevent running zero methane or nitrogen samples as discussed in Section 3.4 below. Either change can be implemented upon request when ordering new analyzers."

Line 195 – Source to the Aeris engineers is missing

The document provided by Aeris Technologies was emailed to the authors. We have added personal communication as the citation.

The findings from (i)&(ii) (Line 183-196) are not very well visible in the graphs, maybe a zoom in to the respective "noise sections" would help.

The data for the other two instruments is so linear that we just increased the y axis to include the full scale of Aeris data.

Line 208 – remove "of"

Done.

Line 215 – Description of Table 2: describe the meaning of "Slope +/-, Intercept +/- and r2".

We have clarified the caption and added 95% CI to the headers in the Table

Table 2. Calibration span (slope) and zero (intercept) calculated for each instrument reporting at 1 Hz when sampling the NOAA calibration standards. The 95% confidence intervals (CI) for the slope and intercept of an Ordinary Least Squares (OLS) fit are also shown. **The ethane Picarro G2210-i calibration was calculated from the mean of each cylinder measurement (two-point calibration).

Figure 2 – y-axis label is not centered.

Centered.

Table 3 – Add sources of quoted CH4 & C2H6 precision

We clarified the wording:

**Table 3: Summary of various instrument performance metrics. The quoted precisions are from the Product Datasheet for each analyzer except \*Aerodyne Superdual Quoted Precision from Kostinek et al., 2019**

Line 232-243: very good description and discussion

Thanks

Figure 3 – increase the spacing between the upper and lower graph (axis overlap)

We have added more space between the figures

Line 283 – remove "during"

Done.

Line 297-304 – very good discussion

Thanks!

Conclusion and Recommendations: Line 309 – replace "or" with "of"

Done.

Conclusion: This paper is very well written and offers a clear structure to follow. The two main drawbacks lie in the description of the need of this study based on previous research and the conclusion being based on the results of only one experiment.

We thank the reviewer for their helpful comments. We have added more motivation for the study. We believe we have clarified the extent of the ambient sampling that extends the comparison beyond a single experiment.